# Exploring the Impact of Pitch-Coated Pottery on Wine Composition: Metabolomics Characterization of an Ancient Technique

**DOI:** 10.3390/foods14223857

**Published:** 2025-11-11

**Authors:** Clara Abarca-Rivas, Julián Lozano-Castellón, Maria Pérez, Marina Corrado, Anna Vallverdú-Queralt, Andrea Zifferero, Riccardo Chessa, Paul Reynolds, Alessandra Pecci, Rosa M. Lamuela-Raventós

**Affiliations:** 1Polyphenol Research Group, Department of Nutrition, Food Science and Gastronomy, Faculty of Pharmacy and Food Sciences, Institute of Nutrition and Food Safety (INSA-UB), “Antioxidants Naturals: Polifenols”, University of Barcelona, 08028 Barcelona, Spain; claraabarca@ub.edu (C.A.-R.); mariaperez@ub.edu (M.P.); marinacorrado@ub.edu (M.C.); avallverdu@ub.edu (A.V.-Q.); 2CIBER Physiopathology of Obesity and Nutrition (CIBEROBN), Institute of Health Carlos III, 28029 Madrid, Spain; 3Department of History and Cultural Heritage, University of Siena, 53100 Siena, Italy; andrea.zifferero@unisi.it; 4Laboratorio di Archeologia Sperimentale Gli Albori, 58042 Campagnatico, Italy; 5Archaeological and Archaeometric Research Unit (ERAAUB), Institute of Archaeology (IA-UB), ICREA, Universitat de Barcelona, 08001 Barcelona, Spain; paulreynolds@ub.edu; 6Archaeological and Archaeometric Research Unit (ERAAUB), Institute of Archaeology (IA-UB), Institute of Nutrition and Food Safety (INSA-UB), Universitat de Barcelona, 08001 Barcelona, Spain

**Keywords:** amphora, wine, HRMS, multivariate data analysis, foodomics, clay vessel, anthocyanins, coating, pitch, UPLC-MS/MS

## Abstract

In recent years, wine producers have increasingly experimented with ancient fermentation and ageing techniques, such as the use of ceramic containers or pitch-coated amphorae. Despite growing interest in these traditional practices, few studies have investigated the chemical composition of the resulting wines. This is the first study that characterises pine pitch, historically used as a coating material, and evaluates the impact of pitched pottery vessels on the winemaking process and wine composition using a foodomics approach. Vinification was carried out in both pitch-coated and uncoated (control) clay containers. Chemical differences between must and wine produced in pitched and unpitched vessels were assessed using targeted and untargeted ultra-performance liquid chromatography–tandem mass spectrometry (UPLC-MS/MS). Compared to the control, must and wine from the pitch-coated vessels had higher concentrations of many polyphenols such as anthocyanins, coumaric acid and tartaric acid, while procyanidins were present in significantly lower amounts. These findings reveal that pitch-coated ceramic vessels significantly influence wine composition, offering a first step toward deeper investigations into how fermentation environments shape metabolite profiles. This knowledge not only enhances our understanding of traditional practices but also opens new avenues for innovation in contemporary oenology.

## 1. Introduction

The revival of traditional winemaking methods is a growing trend in the wine industry, aimed at producing distinctive, higher-quality wines with enhanced regional typicity [1,2,3]. It is well documented that some of the earliest winemaking techniques made use of ceramic vessels [4,5]. In this context, winemakers in wine-producing countries such as France, Portugal, Croatia, USA, Slovenia, Austria, and Italy have been experimenting with fermentation or ageing in amphorae [6]. Studies on the use of ceramic vessels in vinification have focused primarily on wine ageing in modern pottery vessels compared to other materials such as wood or glass but little is known about impact of ancient coated ceramic vessels on the fermentation, overall quality, chemical profile, and sensory characteristics of wine [7,8,9,10]. Although ceramic vessels are frequently referred to as “amphorae” in contemporary winemaking literature, this terminology is archaeologically inaccurate. The term “amphora” specifically denotes an ancient vessel with a pointed base, narrow neck, and typically two handles, designed for the transport and storage of foodstuffs, including wine, but not for the winemaking process itself [11]. Pottery vessels usually used to produce wine were called dolia for the Roman period or pithoi for earlier periods. Dolia have been found in many different archaeological sites devoted to the production of wine in Roman times [12]. Nevertheless, to maintain consistency with current winemaking terminology, we adopt the term “amphora” here to refer to ceramic vessels used in winemaking.

In ancient oenological practices, wine was fermented and aged in clay vessels [13,14]. The porosity of clay facilitates micro-oxygenation of wine, which may contribute to color stabilization [3,15]. However, due to their high permeability, ceramic vessels historically required a waterproofing treatment to hold liquids more effectively [16]. In the Western Mediterranean, one of the most widely used coatings was pitch, a natural organic substance derived from the combustion of plant resins [17,18,19].

The analytical characterisation of pitch remains challenging due to its complex and heterogeneous composition, which includes a wide spectrum of compounds with diverse molecular weights and solubility profiles [20,21]. Gas chromatography coupled with mass spectrometry (GC/MS) has been widely employed for pitch analysis [17,18,22,23], but this technique requires hydrolysis or thermal degradation, potentially compromising structural information. To overcome this, microwave-assisted extraction combined with FIA-ESI-Q-ToF-MS has been proposed, offering improved detection of intact high molecular weight compounds [21]. However, this method still involves elevated temperatures (80 °C) in sample preparation and lacks chromatographic separation, limiting isomer resolution, and complicating structural assignments.

The aim of the present study was to comprehensively characterise traditionally produced pitch using high resolution mass spectrometry (HRMS) and GC-MS and evaluate its impact on wine quality and the winemaking process when applied as an amphora coating. To date, research in this specific field is limited to two studies examining traditional commercial Portuguese Talha wine fermented with no additives in pitch-coated vessels, one focusing on volatile compounds [24] and the other on metal migration from ceramics into the wine [25]. Our work significantly expands on previous research by characterising traditional pitch formulation, assessing the effect of amphora coating on wine composition by comparison with uncoated controls, and investigating possible pitch migration into the wine. Advanced high-resolution and foodomics techniques were used to achieve these goals.

In summary, the objectives of this study are to comprehensively elucidate the chemical profile of traditional pitch, and to evaluate the impact of pitch-coated amphorae on wine composition during vinification through a comparative, non-targeted metabolomics approach.

## 2. Materials and Methods

### 2.1. Chemicals

Acetonitrile, formic acid, hexane, methanol, chloroform and ethanol were purchased from Merck (Darmstadt, Germany). All solvents were of HPLC grade. Ultrapure water was obtained from a Milli-Q water purification system (Millipore, Bedford, MA, USA). Dehydroabietic acid, potassium hydroxide and *N*,*O*-bis(-trimethylsilyl)trifluoroacetamide (BSTFA) were purchased from Sigma-Aldrich (St. Louis, MO, USA). Methyl dehydroabietate, cyanidin 3-*O*-glucoside chloride, malvidin-3-*O*-galactoside chloride, malvidin-3-*O*-glucoside chloride, pelargonidin-3-*O*-glucoside chloride, pelargonidin-3-*O*-galactoside chloride, and petunidin-3-*O*-glucoside chloride were provided by Cymitquimica (Barcelona, Spain). trans-Resveratrol was acquired from Cayman Chemicals (Ann Arbor, MI, USA). All standards were handled under UV-filtered light.

### 2.2. Pitch and Ceramic Vessel Production

Pitch was produced by the Asociación Cabaña Real de Carreteros (RCC) in Quintanar de la Sierra, Burgos (Spain) in a traditional kiln and following historical methods, that involve the burning of resin-rich pieces of *Pinaceae* wood. To remove impurities, 180–200 g batches of pitch were heated over a direct flame for 20 min and then filtered through a sieve to eliminate solid residues. Samples were taken at this stage for pitch chemical characterisation.

The ceramic vessels were made at the Laboratorio di Archeologia Sperimentale “Gli Albori” (Campagnatico, Grosseto, Italy) using ancient techniques in the framework of a long-term international collaboration between the University of Siena and the University of Barcelona. The vessels were produced in one batch using clay sourced from Albinia (Orbetello, Grosseto, Italy) mixed with volcanic temper obtained from Torniella (Roccastrada, Grosseto, Italy) and were subjected to standardised firing process at around 1000 °C. Although surface porosity was not quantitatively measured, the controlled production process yielded consistent ceramic material across the amphorae.

Exceptionally well preserved Roman dolia and amphorae from underwater sites such as the Bay of Cadiz [26] have revealed that pitch coating typically ranged from one millimeter to one centimeter in thickness. In line with this evidence, a pitch layer of about 2 mm was applied to the ceramic vessels used in the experiment. To apply the pitch coating, the vessels were first heated over a direct flame for 20 min. Separately, the pitch was melted over a gas burner until liquified and 0.5 kg of pitch was applied to each vessel. The coating was applied evenly, resulting in a uniform layer across the entire internal surface. Once coated, the vessels were allowed to cool completely prior to filling them with grape must.

### 2.3. Winemaking Using Ceramic Vessels

Experimental fermentations were conducted at the Laboratorio di Archeologia Sperimentale “Gli Albori” in Campagnatico (Grosseto, Italy). A total of 52.5 kg of ripe organically grown Sangiovese red grapes were harvested for vinification, which was carried out using ancient methods (Figure 1). Sangiovese grapes were selected due to their historical relevance in central Italy and availability during the harvest period, ensuring traceability and consistency across experimental conditions. To minimise variability, all grapes were harvested and processed at once. The grapes were manually crushed in a single vat to obtain a homogeneous must, which was then distributed into two types of ceramic vessels—pitch-coated and uncoated (control). This approach ensured that any differences observed could be attributed to the vessel type rather than to differences in grape composition or processing conditions. Alcoholic fermentation (AF) was spontaneous, initiated by native yeasts present on the grape skins and/or in the environment. No commercial yeast strains were added during the AF. Grape skins remained in contact with the must throughout fermentation. The temperature was maintained at 25 °C, and AF was considered complete when the residual sugar concentration fell below 1 g/L. During the 10-day AF period, the must was manually stirred twice a day with a wooden pole. Samples were collected at two time points: at the beginning and at the end of alcoholic fermentation (AF). In both cases, samples were filtered through a stainless-steel mesh to remove grape residues and suspended particles prior to collection All samples were stored in an ultra-freezer at −60 °C until chemical analyses were performed.

### 2.4. Sample Preparation

For pitch characterisation, two different extraction procedures were carried out. For the untargeted metabolomics analysis, 50 mg of pitch was extracted in triplicate using 5 mL of two different solvents: hexane and ethanol. Preliminary testing of solvent extraction using hexane, ethanol, and tert-butyl methyl ether (TBME) were carried out to identify the most effective for analyte recovery and subsequent UHPLC-Q-ToF-MS analysis. Hexane and ethanol showed superior efficiency compared to TBME allowing extraction of non-polar and polar compounds, respectively. The samples were sonicated in an ice bath for 25 min. Although the temperature was not continuously monitored with a probe, maintained at sufficient ice levels to ensure the temperature remained below 10 °C throughout the extraction. Subsequently, 1 mL of each extract was transferred to a glass tube and evaporated to dryness under a gentle stream of nitrogen. The dried extracts were reconstituted in 1 mL of tert-butyl methyl ether, filtered through a 0.2 µm polytetrafluoroethylene filter, and stored in 2 mL amber-glass vials at −80 °C until analysis.

For the GC-MS analysis, lipid extraction was performed on 0.5 g of pitch following established protocols to obtain the so called “total lipid extract” (TLE) (Charters et al. [27]). Briefly, the pitch sample was extracted with 3 mL of CHCl_3_/MeOH (2:1 *v*/*v*), vortexing and then sonicating in an ice bath for 40 min avoiding sample heating over 10 °C. After centrifugation the supernatant was transferred to a 2 mL vial and evaporated under a gentle stream of nitrogen. An aliquot of the TLE (1/4) was trimethylsilylated using *N*,*O*-bis(trimethylsilyl) trifluoroacetamide (BSTFA, 30 μL). μL was injected for GC-MS analysis.

Wine samples were diluted 1/10 with Milli-Q water, filtered through a 0.2 µm nylon membrane, and transferred to amber-glass vials containing 300 microliter inserts. Samples were stored at −80 °C until analysis.

### 2.5. Foodomics Profiling by UHPLC-Q-ToF-MS

Non-targeted metabolomics analysis was performed using an Agilent 6560 Ion Mobility Q-ToF LC/MS system (Santa Clara, CA, USA) working in both positive and negative modes at the Separation Techniques Unit of the Scientific and Technological Centers (CCiTUB), Universitat de Barcelona. Chromatographic separation was achieved using an AcquityTM Ultra Performance Liquid Chromatography (UPLC)^®^ BEH C18 column (2.1 × 100 mm, i.d., 1.7 µm particle size) (Waters Corporation, Ireland), maintained at 40 °C throughout the analysis. The sample sequence was randomised and pooled quality control samples were injected at the beginning of the sequence and every ten runs. The QC pool was prepared by mixing equal aliquots from all individual samples included in the study, ensuring a representative composite for monitoring analytical performance. Internal standards were not included, as the aim of the study was not absolute quantification but rather comparative analysis between samples areas. Data were acquired in data-dependent MS/MS mode using 12 precursors per cycle (1 Hz, 50–1200 *m*/*z*, positive polarity, active exclusion after two spectra), with collision energies of 10, 20 and 40 eV for collision-induced decomposition. The separation methods differed depending on the focus of each analysis: wine metabolomic profiling, confirmatory analysis of wine anthocyanins, and pitch lipidomic characterisation. To evaluate the stability and reproducibility of the analysis, the relative standard deviation (RSD%) of detected features in QC samples was calculated. The distribution of RSD% values is shown in Appendix A, where the majority of features exhibit low RSD%, confirming the robustness and consistency of the analytical platform. Additionally, mass accuracy (in ppm) and signal stability were monitored throughout the sequence to ensure consistent instrument performance.

In the Q-ToF analyses, compounds were putatively annotated (level 2 identification), according to the COSMOS Metabolomics Standards Initiative [28]. Post-acquisition data analysis by UHPLC/Q-ToF was performed using MS-Dial (version 4.9) [29] and MS-Finder (version 3.6) [30]. MS-DIAL was used for automatic peak detection, LOWESS normalization, and compound annotation through spectral matching. The spectral comparison was performed against the MS-DIAL ESI(+)-MS/MS (16,232 unique compounds) and ESI(−) (8887 unique compounds) databases derived from authentic standards. The analysis focused on the *m*/*z* range of 100–1800, with a minimum peak height threshold of 1000 counts per second (cps) for both polarities. The tolerance levels for peak centroiding were set at 0.01 Da for MS and 0.05 Da for MS/MS. Retention time data was not included in the scoring process. For compound identification, tolerances of 0.01 Da for MS and 0.05 Da for MS/MS were applied, considering mass accuracy, isotopic pattern, and spectral similarity. These criteria contributed to a comprehensive total identification score within MS-DIAL, with a cutoff threshold of 75% similarity according to the accepted standards in untargeted metabolomics and aligned with common ion adducts in lipidomics and metabolomics. To address missing peaks, a gap-filling algorithm with a 5 ppm *m*/*z* tolerance was employed.

Unannotated features were further analysed using MS-Finder [30] for in silico fragmentation. Searches were conducted against multiple databases, including LipidMaps, YMDB, PlantCyc, ChEBI, NPA, NANPDB, COCONUT, UNPD, KNApSAcK and FoodDB. Only compounds achieving an in silico prediction score above 5 were retained for further analysis.

The following chromatographic gradients were performed with the same column and conditions previously described in this section.

#### 2.5.1. Untargeted Analysis of Wine

Mobile phases were water (A) and acetonitrile (B), both containing 0.1% formic acid. The solvent gradient (*v*/*v*) of B (t (min), %B) was defined as follows: (0, 2); (2, 2); (4, 30); (8, 100); (10, 100); (11, 2); and (14, 2). The injection volume was 4 μL and the flow rate was set at 0.4 mL/min.

#### 2.5.2. Wine Anthocyanins

For the targeted analysis of anthocyanins, the mobile phases were water (A) and acetonitrile (B), both containing 5% formic acid. The solvent gradient (*v*/*v*) of B (t (min), %B) was defined as follows: (0, 2); (3, 2); (5.5, 5); (6.5, 7); (13, 12); (15, 80); (18, 80); (18.1, 2); and (20, 2). The injection volume was 4 μL, with a flow rate of 0.4 mL/min.

#### 2.5.3. Pitch Lipidomics

For lipidomics analysis of the pitch extracts, the mobile phases consisted of (A) 5 mM ammonium formate and 0.1% formic acid in water/methanol (95/5, *v*/*v*), and (B) 5 mM ammonium formate and 0.1% formic acid in 2-propanol/methanol/water (65/30/5, *v*/*v*/*v*). The linear gradient and flow rate increased linearly as follows, considering time (min), %B, flow rate (µL/min): (0, 10, 200), (5, 50, 200), (15, 80, 250), (28, 100, 250), (30, 100, 250), (30.9, 10, 250), and (35, 10, 250). The injection volume was 5 μL.

### 2.6. Semi-Volatile Organic Compound Analysis by GC-MS

GC-MS analyses were carried out at the GC Unit of the Scientific and Technological Centers (CCiTUB), Universitat de Barcelona. The instrumentation consisted of an Agilent 6890N gas chromatograph (Agilent Technologies, Santa Clara, CA, USA) equipped with a silica capillary column Rtx-1 (60 m × 0.32 mm × 0.1 μm) and a 5975 inert XL mass spectrometer operated in full scan mode (*m*/*z* 40–650). Samples were introduced via an Agilent 7683BSeries autoinjector (Agilent Technologies), and electronic ionization was conducted at 70 eV. The GC oven temperature was held at 50 °C for 1 min, then increased at 5 °C/min up to 300 °C and held isothermally for 10 min. Peaks identification was carried out by comparing spectra with the NIST (version 2.0) and published spectra.

### 2.7. Anthocyanin Quantification by UPLC-DAD

Anthocyanins were initially identified by LC/MS/MS and subsequently quantified by an ultra-performance liquid chromatography-diode array detector (UPLC-DAD), following the method described by Sandoval et al. [31]. A Waters Acquity UPLC H-class system (Waters Corp, Milford, MA, USA) coupled to a photodiode array (PDA) detector (Waters Corp, Milford, MA, USA) with a BEH C18 column (50 mm × 2.1 mm) i.d., 1.7 µm (Waters, Milford, MA, USA) was used. The injection volume was 10 µL. Samples were maintained at 4 °C and the column at 30 °C. The mobile phase consisted of an A phase of acetonitrile (5% formic acid), and a B phase of water (5% formic acid). A gradient elution was performed with the following time-based composition: at 0 min, the mobile phase was 2% A and 98% B, which was held for 3 min. The gradient progressed to 5% A and 95% B at 5.5 min, 7% A and 93% B at 6.5 min, and continued to increase to 12% A and 88% B at 13 min. At 15 min, the gradient reached 80% A and 20% B, which was maintained until 18 min. The gradient returned to 2% A and 98% B at 18.1 min, and this composition was sustained until the end of the run at 20 min.

Identification was achieved by comparing the retention time of the chromatographic peaks with the those of the pure standards. In cases where standards were unavailable (delphinidin-3-*O*-glucoside and peonidin-3-*O*-glucoside), identification was supported by HRMS, as described previously. Anthocyanins were quantified using calibration curves with standards of cyanidin-3-*O*-glucoside, petunidin-3-*O*-glucoside, and malvidin-3-*O*-glucoside. The rest of the anthocyanins were quantified using the compound that is most similar in structure.

### 2.8. Multivariate Date Analysis

A chemometric approach using multivariate statistics was employed to analyse the foodomics data. The raw dataset used was exported to either SIMCA software v13.0.3.0 (Umetrics, Sweden) or Metaboanalyst 5.0 [32].

SIMCA was used for unsupervised analysis. Following Lozano-Castellón et al. [33], the data were standardised using unit-variance (UV) scaling, applying the formula: (value − mean)/standard deviation. To determine which variables varied significantly with the pitch coating, a supervised orthogonal partial least squares-discriminant analysis (OPLS-DA) was performed, using pitch coating as the discriminating factor. This resulted in two classes within the Y matrix. The X matrix consisted of the peak areas of the compounds. For this model, UV scaling was applied again, and variables were log-transformed when necessary, using the SIMCA auto-transform option. Variables with a variable importance in projection (VIP) score greater than 1.5 were identified as significant markers.

To validate the model, we assessed goodness-of-fit (R^2^Y) and goodness-of-prediction (Q^2^Y), with a Q^2^Y threshold of >0.5 indicating acceptable predictability. Hotelling’s T^2^ statistic was employed to identify potential outliers, using 95% and 99% confidence intervals to classify suspicious and strong outliers, respectively. Additionally, potential outliers were evaluated using a residuals normal probability plot (see Appendix A). The model was cross-validated, and an ANOVA of the cross-validated residuals was performed to test if the model was fitted by chance, using a significance threshold of *p* < 0.01. Lastly, a permutation test with 200 permutations was conducted to further assess and rule out the risk of overfitting.

The subsequent analyses were performed using Metaboanalyst 5.0 [32]. To identify marker compounds in each scenario, Student’s *t*-tests were used to assess significant differences between groups. Initially, the data were prepared for testing through auto-scaling. To minimize the risk of a type I error, the false discovery rate was applied, setting the threshold at 0.05.

### 2.9. Monovariate Statistics

The results are expressed as means ± standard deviations. All samples were analysed three times. Statistical significance was determined using Student’s *t*-test at a 95% confidence level (*p*-value < 0.05). The statistical analyses to evaluate differences between anthocyanin content in all the wine samples and during the winemaking process were conducted using STATA software (version 16.0; StataCorp, College Station, TX, USA).

## 3. Results and Discussion

### 3.1. Pitch Characterisation

As previously noted, pitch was traditionally used to line Mediterranean amphorae and *dolia* and was produced through the strong heating and distillation of *Pinaceae* resins. To determine its composition, a GC-MS method usually used to characterize organic residues in archaeology was applied to analyse pitch samples. Several compounds recognised as pitch biomarkers were identified, including dehydroabietic acid (DHA) and 7-oxo-dehydroabietic acid (DHA), which are characteristic of *Pinaceae* resins. Retene, a biomarker indicative of resin exposure to high temperatures [34,35], was also detected, as was methyl dehydroabietate (MDHA), which indicates that pitch was produced via wood distillation [35,36]. Notably, DHA is known for its broad biological activity, including antimicrobial properties [37,38].

To further characterise the pitch, an untargeted metabolomics analysis was performed using UHPLC-QToF-HRMS, leading to the putative identification of 30 compounds (Table 1). The major constituents were diterpenes derived from abietic acid, a characteristic compound of pine resins (Figure 2).

Briefly, in the presence of oxygen, abietic acid undergoes allylic oxidation at the C-ring, resulting in the generation of 12-oxo-abietic acid, an oxidation product identified in vitro [39,40]. Through isomerisation and aromatisation, abietic acid is converted to DHA, its most common metabolite [36,41]. DHA can undergo oxidative decarboxylation to form 18-nor-abietatriene or benzylic oxidation to yield 7-hydroxydehydroabietic acid, both of which have been reported previously in pitch [34,42]. Additionally, dehydroabietinal is transformed into DHA by oxidation [43].

Some studies have shown that certain oxidation products of resin acids, such as methyl 7-oxodehydroabietate and methyl 7-hydroxydehydroabietic acid, may cause allergenic reactions [40,44]. As the latter compound was detected in the pitch, its potential allergenicity warrants investigation into whether it can migrate into wine. Another diterpene identified in the pitch (compound **10**, Table 1) is a derivative of beta-phellandrene, a constituent of pine resin [45,46]. Additionally, phenanthrene carboxylic acid (Table 1), classified as a polycyclic aromatic hydrocarbon (PAH), was detected. There is growing evidence that PAHs, even at low levels of exposure, can affect the immune system and humoral responses [47,48]. Given the potential toxicity of these compounds, it is essential to assess their possible migration into food or beverages stored in pitched-coated ceramic vessels.

Among the sesquiterpenes, beta-ionone was putatively identified (Table 1), likely derived from alpha- or beta-pinene through a series of enzymatic oxidation and rearrangement reactions [49]. Beta-ionone is commonly associated with a violet-like aroma [50]. Additionally, farnesal, a compound naturally present in pine nuts [51], was detected (Table 1). Farnesal has been shown to have antimicrobial [52] and antifungal activities, which may help to improve wine preservation [53].

Several compounds resulting from the pitch extraction process were also identified, including oleamide and palmitic amide (Table 1), both of which can form during the high-temperature heating of pine wood [54]. In addition, naphthalene derivatives such as compound **23** and **24** (Table 1) could be generated during this process. Some compounds of this category have been detected in amphorae coated with pine pitch [45]. Moreover, certain naphthalene derivatives exhibit antimicrobial activity, with studies reporting inhibitory effects against pathogens such as *Staphylococcus aureus* or *Escherichia coli* [55]. These findings support the hypothesis that pitch may have served not only as a waterproofing sealant but also as a preservative in ancient food and wine storage [35].

Another group of compounds identified in the pitch with potential preservative properties includes stilbenes and other aromatic compounds associated with vanillin (Table 1). For example, dimethoxystilbene has been reported as an intermediate in the biosynthesis of pinosylvin, a polyphenol known for its antifungal, antibacterial, and other biological activities [56,57]. The vanillin-related aromatic hydrocarbons detected in the pitch are likely products of the thermal degradation of lignin and may also arise from the methylation of lignin-derived acids during pyrolysis [58,59]. Both pinosylvin-related stilbenes and vanillin derivatives have demonstrated promising antioxidant properties [60,61].

Finally, two alkaloids—pinidine and pinidinol—previously reported in various Pinus species, were putatively identified in the pitch (Table 1). Pinene-derived alkaloids exhibit a broad range of health-related properties [62]. For instance, 1,6-dehydropinidine exhibits antibacterial activity against multiple bacterial strains, suggesting its potential as a natural antimicrobial agent [63]. Moreover, computational analyses revealed that pinidine may serve as a promising PARP inhibitor in the context of pancreatic cancer [64].

### 3.2. Effect of Pitch on Winemaking

A total of 69 compounds were putatively identified in the experimental must and wine through untargeted metabolomic analysis (Table 2). The specific distribution of these compounds across sample types: must and wine from pitch-coated and un-coated ceramic vessels is detailed in Appendix A, which includes the peak area data for all samples. Importantly, none of the potentially hazardous compounds identified in the pitch resin (Table 1) were detected in the experimental wine, suggesting minimal or no migration from the coating into the final product. The absence of compound transfer may be due to the hydrophobic nature of the pitch [65]. Only ethyl vanillin was detected in the resin as well as in wines fermented in both pitch-coated and uncoated amphorae. Since it was also present in the control vessels, its origin is unlikely to be solely attributable to the pitch coating. This compound is known to occur naturally in wine as a result of fermentation [66], supporting the interpretation that its presence is fermentation-derived rather than pitch-derived. However, as this study was based on a non-targeted metabolomics approach, specific detection limits for individual hazardous compounds such as PAHs were not established. Therefore, while no pitch-derived compounds were detected under the conditions used, we note that trace-level migration cannot be entirely ruled out without targeted analysis using authentic standards. Moreover, it is necessary to clarify if, in the long-term contact with pitch-coated vessels (e.g., ageing), there is any migration into the wine.

#### 3.2.1. Wine and Must Chemical Profile

To investigate the effect of pitch coating on vinification, an OPLS-DA was performed on all samples. Figure 3 shows the OPLS-DA scores plot, with samples colored and shaped according to the amphora type (presence/absence of coating) and vinification stage. The validation parameters of the supervised model were as follows: R2Y(cum) = 0.934, Q2(cum) = 0.85, and *p*-value Cross Validation-ANOVA = 3.99 × 10^−15^. Detailed validation results are provided in Appendix A. The score plot revealed a clear separation of samples by coating status and winemaking stage. Principal Component 1 (PC1) primarily accounted for the discrimination between coated and uncoated amphorae and vinification stages, while Principal Component 2 (PC2) differentiated must from wine samples.

The marker compounds associated with the amphora pitch coating are listed in Table 3, along with their VIP scores, ANOVA or *t*-test *p*-values, and the values of each variable for every level of the corresponding factor. Among these, coumaric acid, esculetin, tartaric acid and procyanidin C1 dimer 1, 2 and 3, procyanidin B1 dimer 2 and kynurenic acid had the highest VIP scores (>1.16), indicating a strong contribution to the model. Of these compounds, all were increased in the wine produced in pitch-coated amphorae compared to non-coated except for procyanidins. Esculetin and procyanidins are derived from the general phenylpropanoid biosynthetic route, which is closely regulated by yeast metabolism [67,68]. In wines fermented in pitch-coated amphorae, elevated levels of phenylalanine, esculetin, and flavonoids such as quercetin were observed. These increases may be attributed to the reduced micro-oxygenation provided by the pitch, which favors the accumulation of oxidation-sensitive compounds [69,70]. Moreover, the presence of the resin may influence enzymatic activity of yeast or redirect metabolic fluxes toward the biosynthesis of flavonols (e.g., quercetin) and coumarins (e.g., esculetin), limiting the availability of catechin for procyanidin assembly [68]. This metabolic reprogramming could account for the observed decrease in procyanidin levels.

Furthermore, coumaric acid, one of the most significant hydroxycinnamic acids in wine, plays a key role in color stabilization, aroma development, and antioxidant capacity [71,72]. Its enrichment in pitch-coated amphora wines is likely due to the limited oxygen exposure, which reduces oxidative degradation—a process to which coumaric acid is particularly sensitive [73,74,75]. Lower oxygen permeability helps preserve coumaric acid and its derivatives, enhancing copigmentation with anthocyanins and contributing to color intensity, especially in young red wines [76].

In addition, tartaric acid emerged as a prominent compound in wines fermented in pitch-coated amphorae, with concentrations significantly higher than in the control wine. Elevated tartaric acid levels are associated with increased acidity [77]. One plausible explanation is that wines fermented in uncoated vessels may experience greater tartrate precipitation due to higher oxygen permeability and surface interactions, which promote polymerization of tartaric acid with polyphenols and other wine constituents [77,78]. This precipitation reduces the amount of tartaric acid remaining in solution, potentially explaining the lower concentrations observed in the control wine.

Finally, kynurenic acid, identified as a VIP marker in the model, was also higher in pitch-coated wine. Its concentration depends primarily on tryptophan availability and specific yeast strain activity [79]. The lower levels of kynurenic acid in wines from uncoated amphorae were likely due to reduced conversion of tryptophan, which was present in higher concentrations in these samples. Notably, kynurenic acid may modulate alcohol metabolism by inhibiting aldehyde dehydrogenase activity [80].

In conclusion, these changes during alcoholic fermentation suggest that *Pinaceae* resin coating not only influences oxygen permeability but also modulates key metabolic pathways during fermentation. Future studies involving targeted enzymatic assays, microbial profiling, and transcriptomic analyses will be necessary to confirm the underlying mechanisms.

#### 3.2.2. Anthocyanins

Anthocyanins in the experimental must and wine samples were identified by UHPLC-Q-ToF-MS and quantified by UPLC-DAD, employing the following standards: cyanidin-3-*O*-glucoside, petunidin-3-*O*-glucoside, and malvidin-3-*O*-glucoside (Appendix A). Delphinidin-3-*O*-glucoside and peonidin-3-*O*-glucoside were putatively identified through non-targeted UHPLC-Q-ToF-MS analysis. All identified anthocyanins have been previously reported in Sangiovese wines [81,82].

Table 4 presents the average concentration of each anthocyanin across the samples. Malvidin-3-*O*-glucoside was the most abundant pigment in all samples, accounting for over 50% of the total quantified anthocyanins, consistent with previous findings in Sangiovese wines from Crimea [83]. As shown in Table 4, most anthocyanins decreased during the winemaking process. This decline was likely due to the formation of more stable pigment forms over time such as pyranoanthocyanins, that have been identified in wine-model solutions [76,84,85].

The differences in anthocyanin content between the two types of wine were evident not only in the absolute values but also in its evolution throughout the vinification process, as shown in Table 4. Wines produced in pitch-coated amphorae consistently contained higher concentrations of all quantified anthocyanins. As reported by Nevares and del Alamo-Sanza, when comparing uncoated clay vessels to those coated with beeswax mixed with colophony or almond oil [3], the oxygen transmission rate was found to be higher in non-coated amphorae, which could explain the higher oxidation of anthocyanins in these samples. Additionally, pigment losses in uncoated vessels may have been exacerbated by evaporation and diffusion through the porous ceramic walls, particularly during early fermentation.

## 4. Conclusions

This study presents, to the best of our knowledge, the first comprehensive characterisation of the impact of *Pinaceae* pine pitch amphorae coating on wine production employing high resolution metabolomics techniques. The analysis revealed that pitch coating contains several antimicrobial metabolites with potential preservative properties. No direct transfer of pitch-derived compounds into the wine was detected. Despite the absence of direct migration, pitch coating was found to significantly affect wine chemistry, altering the concentrations of key metabolites such as flavanols and organic acids. Multivariate analysis of the data confirmed that both amphora coating is a determinant of the chemical profile of wine. Compared to uncoated controls, wines produced in pitched-coated amphorae exhibited higher levels of some polyphenols such as anthocyanins or coumaric acid while procyanidins were present in lower amounts. These findings suggest that pitch-coated ceramics modulate wine composition in meaningful ways, providing new insights into a winemaking technique rooted in Roman antiquity. These results represent a first step toward understanding how pitch coating can modify the fermentative environment and the wine’s metabolite profile. However, further mechanistic studies will be necessary to confirm the underlying processes and to assess the implications of these changes in wine quality and its organoleptic characteristics. This knowledge contributes to our understanding of traditional winemaking practices and may inform future research directions. However, any potential applications in contemporary oenology should be considered exploratory at this stage, as further investigation is needed to evaluate the scalability, reproducibility, and practical feasibility of pitch-coated vessels in modern production contexts.

## Figures and Tables

**Figure 1 foods-14-03857-f001:**
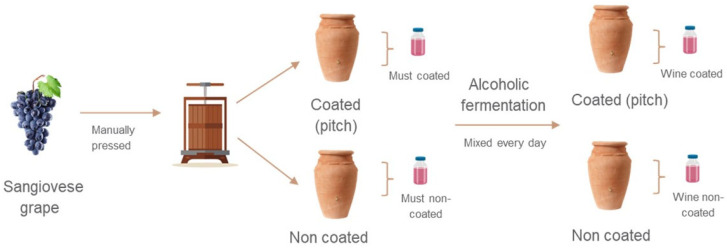
Experimental winemaking process and sampling.

**Figure 2 foods-14-03857-f002:**
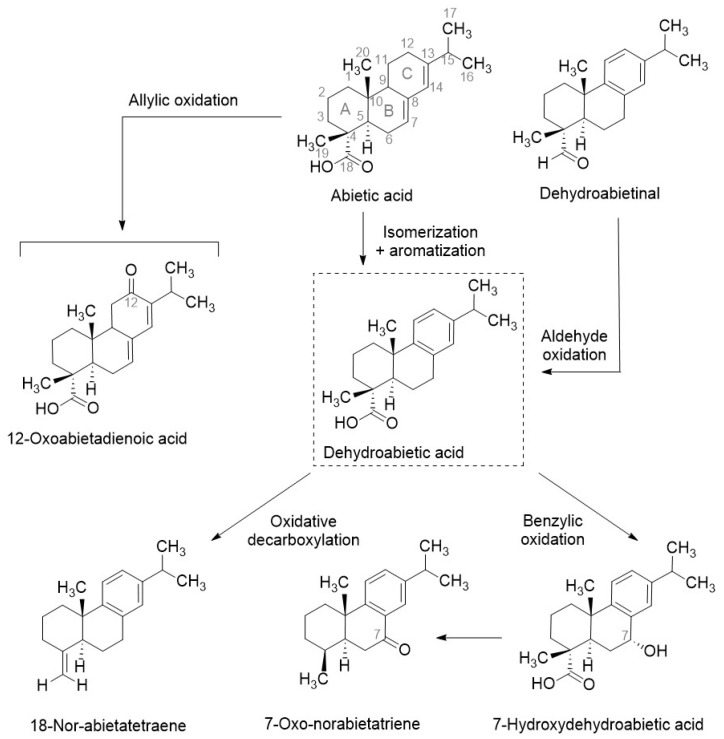
Hypothesised metabolic pathways involving abietic acid and its metabolites in pine pitch, based on annotated compounds.

**Figure 3 foods-14-03857-f003:**
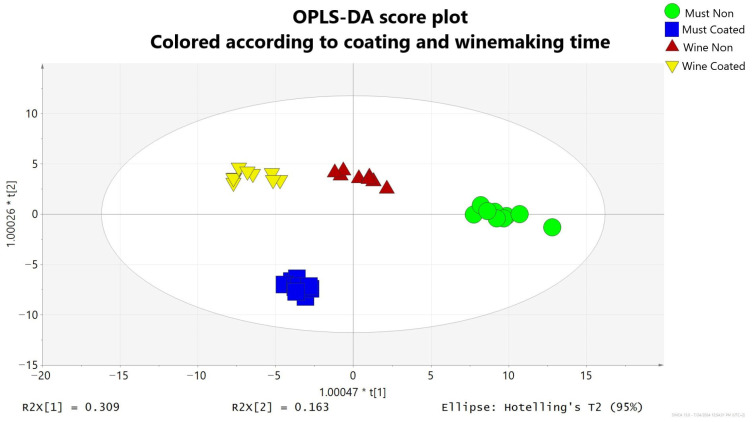
OPLS-DA score plot for the must and wine samples colored according to winemaking time and coating. Must Non = must produced in non-coated amphorae, Must Coated = must produced in pitch-coated amphorae; Wine Non = wine fermented in non-coated amphorae; Wine coated = wine fermented in pitch-coated amphorae.

**Table 1 foods-14-03857-t001:** Compounds tentatively identified in pitch extracts by ultra-high performance liquid chromatography coupled with ion mobility quadrupole time-of-flight mass spectrometry (UHPLC-Q-ToF-MS).

Metabolite Name	Exact Mass	Adduct	RT (min)	Error (ppm)	MS/MS Majority Fragments (Intensity)
Diterpenoids					
Hydroxydehydroabietic acid isomer 1	299.20145	[M−H_2_O+H]^+^	7.187	−2.7	155.08615 (16,282); 159.11694 (5873); 211.14789 (6354); 253.19463 (11,701); 197.13223 (6035)
Hydroxydehydroabietic acid isomer 2	299.20139	[M−H_2_O+H]^+^	7.908	−2.7	155.08594 (16,919); 253.19516 (12,193); 211.14859 (9890); 133.10124 (8078); 117.07057 (5019)
Hydroxydehydroabietic acid isomer 3	299.20132	[M−H_2_O+H]^+^	12.104	−2.3	155.08569: (29,841); 253.19525: (27,501); 211.14841 (17,932); 197.13234 (14,186); 133.10144 (12,894)
Abietic acid	303.23254	[M+H]^+^	8.214	−2.1	123.11738 (25,683); 121.1014 (17,131); 135.11754 (14,543); 107.08598 (13,703); 149.13266 (9772)
Nor-abietatetraene isomer 1	255.21158	[M+H]^+^	9.418	−3.4	171.11717 (38,139); 143.08554 (9124); 156.09375 (6957); 128.06276 (2680); 142.07762 (2179)
Nor-abietatetraene isomer 2	255.21191	[M+H]^+^	9.746	−4.6	171.11713 (14,598); 143.08574 (3778); 123.11701 (2977); 199.14777:2267; 156.09427 (2205)
Dehydroabietic acid isomer 1	301.21686	[M+H]^+^	11.935	−2.3	199.14795 (10,474); 109.10152 (10,150); 143.08582 (9216); 105.07001 (9042); 121.10129 (8698)
12-Oxo-abietic acid	317.21185	[M+H]^+^	12.055	−2.3	317.21127 (13,759); 107.04906 (6792); 147.08023 (4061); 133.06474 (3628); 59.11662 (2141)
Phenanthrene carboxylic acid dodecahydrohydroxydimethylmethylethyl	317.21194	[M−H_2_O+H]^+^	12.27	−2.1	155.08615 (16,282); 253.19463 (11,701); 197.13223 (6035); 211.14789 (6354); 159.11694 (5873)
7-Oxo-norabietatriene	269.19101	[M+H]^−^	12.871	−3.8	213.12778 (112,433); 171.0808 (106,761); 227.14305 (17,489); 211.11169 (11,307); 143.0858 (10,233)
Dehydroabietic acid isomer 2	301.21704	[M+H]^+^	13.373	−2.7	133.10153 (3843); 105.06971 (2939); 185.1319 (2784); 147.11729 (2639) 143.08575 (2300);
Dehydroabietinal isomer 1	285.22205	[M+H]^+^	13.569	−2.5	145.10132 (21,886); 117.06982 (14,934); 125.09614 (12,849); 133.1012 (12,851); 105.07013 (12,318)
Dehydroabietinal isomer 2	285.22223	[M+H]^+^	13.704	−3.2	145.1013 (23,601); 117.07012 (9694); 125.09616 (8414); 105.06988 (6866); 187.14833 (4514)
Salvirecognine	271.20648	[M+H]^+^	13.853	−3.2	147.08063 (34,785); 109.10129 (13,594); 155.08595 (12,211); 189.12756 (11,643); 271.2059 (11,201)
Sesquiterpenoids					
Beta-Ionone	193.15941	[M+H]^+^	6.138	3.24	107.0854 (616); 137.0943 (535); 175.14847 (500); 133.10007 (416); 178.07945 (404)
Farnesal	221.19019	[M+H]^+^	11.306	−0.9	221.19031 (3248); 178.134 (503); 150.10321 (454); 165.12466 (373); 149.09468 (290)
Fatty amides					
Oleamide	282.27997	[M+H]^+^	7.209	−3	107.08616 (1178); 100.07643 (888); 121.1006 (642); 109.10107 (492); 114.09127 (440)
Palmitic amide	256.26443	[M+H]^+^	8.346	−3.5	256.26373 (15,856); 102.09136 (5345); 116.10683 (1647); 130.12434 (571); 186.1833 (497)
Dodecanamide	200.20161	[M+H]^+^	12.186	−3.6	102.09159 (1735); 116.10857 (868); 200.2011 (6335); 100.07675 (466); 130.12204 (229)
Linoleamide	280.26413	[M+H]^+^	13.047	−2.2	109.10132 (6707); 133.101 (5098); 107.08582 (4150); 105.06998 (3705); 121.10113 (3414)
Docosenamide	338.34256	[M+H]^+^	13.918	−2.5	121.10126 (2464); 100.0754 (1506); 107.08583 (1171); 135.11635 (1093); 114.09082 (997)
Naphthalene derivatives					
Dihydrotrimethylnaphthalene isomer 1	173.13297	[M+H]^+^	7.011	−3	131.08594 (3189); 116.06136 (358); 129.06964 (274); 115.05321 (212); 117.0714 (135)
Dihydrotrimethylnaphthalene isomer 2	173.13316	[M+H]^+^	7.951	−4.2	173.13266 (8316); 129.06953 (3288); 116.06165 (1697); 115.05361 (955); 117.06962 (808)
Vinylphenyl compounds					
Methoxy-phenylethenyl phenol	227.10768	[M+H]^+^	6.794	−4.6	121.06529 (8679); 117.07023 (4178); 103.05476 (3953); 212.0831 (3704); 149.06018 (2917)
Dimethoxystilbene	241.12334	[M+H]^+^	7.558	−4.1	117.07041 (9906); 226.09904 (8910); 194.07274 (5523); 165.07045 (5174); 115.05405 (4725)
Aromatic compounds					
Acetovanillone	167.07111	[M+H]^+^	6.968	−4.9	121.02875 (20,456); 149.05962 (4628); 167.06958 (1376); 139.03828 (408); 149.00977 (257)
Ethyl vanillin	167.07088	[M+H]^+^	12.022	−3.8	121.02848 (61,958); 149.05972 (16,175); 167.07042 (3802); 139.039 (3089); 123.0803 (1574)
Piperidines					
Pinidinol	158.15468	[M+H]^+^	6.226	−4.8	102.09148 (1818); 126.12804 (282); 116.04947 (251); 130.98596 (214); 143.08549 (97)
Pinidine	140.14388	[M+H]^+^	8.63	−3.76	140.144 (72,697); 138.12784 (6136); 111.1044 (3461); 110.09634 (2206); 112.11227 (2156)

**Table 2 foods-14-03857-t002:** Compounds tentatively identified in must and wine samples by ultra-high performance liquid chromatography coupled with ion mobility quadrupole time-of-flight mass spectrometry (UHPLC-Q-ToF-MS).

Metabolite Name	Exact Mass	Adduct	RT (min)	Error (ppm)	MS/MS Majority Fragments (Intensity)
Aminobutyric acid betaine	146.11815	[M+H]^+^	0.62	−3.0	100.07588 (159); 101.05954 (45); 114.09331 (53); 146.12959 (1553)
Tartaric acid	149.00964	[M−H]^−^	1.067	−2.9	103.0052 (123); 105.01963 (235); 132.07819 (90); 148.90419 (98)
Acetylproline	158.08163	[M+H]^+^	3.364	−1.5	112.07526 (467); 116.06914 (393); 117.06858 (124); 158.11858 (58)
Levoglucosan	161.0462	[M−H]^−^	2.851	−4.0	101.02549 (761); 115.00488 (163); 117.05934 (66); 160.89748 (48)
Tryptophol	162.09164	[M+H]^+^	5.192	−2.8	117.07004 (1833); 130.06619 (731); 143.0726 (1651); 144.08101 (16,180)
Coumaric acid	163.04121	[M−H]^−^, [M+H−H_2_O]^+^	4.063	−4.5	113.30286 (78); 119.0507 (3117); 130.0323 (64); 163.00392 (219)
Phenylalanine	166.08661	[M+H]^+^, [M−H]^−^	2.487	−2.7	103.05387 (1210); 107.04926 (395); 120.08129 (5796); 119.05094 (3710)
Ethyl vanillin	167.07057	[M+H]^+^	3.578	−2.6	106.04202 (382); 123.04397 (1151); 124.05251 (358); 167.0713 (2675)
Gallic acid	169.01564	[M−H]^−^	1.599	−5.0	107.01315 (144); 124.01756 (1025); 125.0247 (8539); 161.38852 (219)
Pyridoxine	170.08177	[M+H]^+^	1.583	−2.5	107.01307 (3051); 109.02811 (1721); 125.02496 (820); 153.01802 (960)
Aesculetin	177.02003	[M−H]^−^, [M+H]^+^	4.187	−3.8	105.03596 (1012); 133.03026 (797); 149.024 (259); 177.0195 (2110)
Dihydroxycoumarin	177.02013	[M−H]^−^	3.625	−4.3	129.54765 (190); 133.02939 (113); 162.85072 (92); 169.45374 (58)
Tyrosine	182.08162	[M+H]^+^	0.996	−4.0	119.04949 (1222); 123.04349 (1273); 136.07573 (1754); 147.04497 (514)
*O*-methylgallic acid	185.04503	[M+H]^+^	2.272	−3.0	111.04411 (93); 139.03758 (558); 143.89607 (179); 144.90154 (80)
Tryptophan	188.07155	[M+H]^+^	3.728	−2.1	118.06576 (6121); 143.07353 (1111); 144.08104 (2491); 146.05969 (5360)
Kynurenic Acid	190.04971	[M+H]^+^	3.964	−0.2	107.04875 (99); 116.04951 (709); 144.04475 (4077); 162.21191 (56)
Citrate	191.02113	[M−H]^−^	0.956	−4.5	103.04139 (381); 111.00914 (22,274); 129.01886 (391); 130.54572 (212)
Ferulic acid	193.05124	[M−H]^−^	4.199	−2.4	126.96464 (55); 133.02979 (165); 133.16936 (36); 134.03615 (249)
Carvyl acetate	195.13777	[M+H]^+^	5.746	0.8	109.10175 (583); 119.08447 (283); 121.10174 (414); 137.09581 (1376)
Linalool acetate	195.13969	[M−H]^−^	6.768	−3.2	160.84276 (201); 167.14275 (176); 179.10927 (205); 195.13882 (1264)
Indolelactic acid	206.08124	[M+H]^+^	4.465	−1.6	118.06466 (1050); 130.06508 (557); 170.06226 (377); 188.07065 (371)
Pantothenic acid	218.10405	[M−H]^−^	3.428	−3.2	116.07175 (300); 146.08296 (781); 208.59618 (118); 218.10118 (103)
Coniferyl acetate	221.08205	[M−H]^−^	5.118	−0.3	149.05922 (237); 157.06602 (135); 176.96494 (267); 195.9342 (125)
Leucylproline	229.15515	[M+H]^+^	1.578	−2.3	114.09164 (283); 121.02687 (308); 142.08659 (3633); 229.15413 (3455)
Prolyl-isoleucine	229.1552	[M+H]^+^	0.887	−3.2	114.05534 (2929); 116.07116 (1570); 142.08662 (21,438); 229.15469 (27,800)
Cysteinosuccinic acid	238.03856	[M+H]^+^	0.689	−2.6	102.9845 (2398); 120.99609 (1341); 156.01283 (577); 184.00565 (718)
Prenyl caffeate	249.11253	[M+H]^+^	4.685	−1.5	107.05048 (443); 129.07048 (978); 131.08498 (500); 204.07907 (589)
N-(tetradecanoyl)ethanolamine	272.259	[M+H]^+^	6.142	−2.2	155.85768 (111); 171.8539 (419) 189.86389 (186); 258.27927 (2915)
Catechin	289.07352	[M−H]^−^, [M+H]^+^	4.039	−4.6	109.03095 (5688); 121.0307 (903); 123.04592 (4149); 125.02528 (2574)
Epicatechin	291.08679	[M+H]^+^	4.262	−1.6	123.04404 (8652); 139.03899 (10,130); 147.04361 (2115); 161.05923 (1763)
Argininosuccinic acid	291.13089	[M+H]^+^	0.777	−3.4	116.07103 (1470); 130.09784 (1057); 134.04478 (1109); 158.09224 (1062)
Cyclopenteneoctanoic acid	293.21182	[M+H]^+^	5.667	−2.3	105.06916 (381); 107.0499 (247); 107.08562 (580); 109.09935 (218)
Tetradecyliminodiethanol	302.30579	[M+H]^+^	6.464	−2.5	104.06934 (214); 256.26303 (3755); 302.27063 (586); 302.3046 (460)
Quercetin	303.05035	[M+H]^+^	4.604	−1.5	128.86797 (196); 137.02216 (410); 229.04767 (193); 303.04892 (1683)
Gentisic acid *O*-hexoside	315.07315	[M−H]^−^	3.354	−3.3	108.0223 (12,894); 109.03048 (3425); 152.01173 (7460); 153.02017 (1827)
Coumaric acid *O*-hexoside	325.09305	[M−H]^−^	4.069	−0.6	117.03597 (291); 119.05083 (20,129); 145.0298 (1917); 163.0405 (4063)
Vanilloyl-*O*-hexoside	329.08859	[M−H]^−^	4.415	−2.4	123.0458 (4655); 125.02437 (710); 167.0363 (5833); 191.03442 (568)
Gallic acid-*O*-hexoside	331.06735	[M−H]^−^	2.05	−1.0	125.02487 (1437); 151.00302 (286); 165.01988 (316); 169.01411 (2356)
Pyridoxine-*O*-hexoside	332.13547	[M+H]^+^	0.944	−4.9	108.08124 (27,296); 124.07555 (4617); 134.06006 (2966); 136.07584 (4181)
Methyl gallate-*O*-hexoside	345.08395	[M−H]^−^	2.271	−3.7	139.04128 (13,792); 140.04408 (265); 153.05818 (242); 163.0403 (474)
Carboxymethyl epicatechin	347.07755	[M−H]^−^	4.592	−0.7	121.02943 (307); 166.02495 (274); 271.6481 (197); 291.69943 (171)
6-O-fatty acyl-*O*-hexoside	353.07346	[M−H]^−^	0.834	−2.7	111.00956 (36,441); 112.01301 (284); 154.99751 (322); 165.95815 (173)
Syringic acid-*O*-hexoside	359.09894	[M−H]^−^	3.628	−1.5	123.00919 (1313); 138.03281 (2358); 153.05646 (1122); 182.02304 (1853)
Syringin-*O*-hexoside	359.09952	[M−H]^−^	1.157	−3.1	128.96178 (197); 167.95267 (283); 170.97278 (262); 182.00439 (252)
Tetrahydroxy-tetramethoxyflavone	387.07211	[M−H_2_O−H]^−^	3.935	0.1	125.02521 (394); 164.01118 (951); 167.03485 (511); 308.02902 (215)
Tetrahydroxy-dimethoxyflavone-acetate	387.07275	[M−H]^−^	4.112	0.9	125.02503 (959); 163.00285 (487); 164.01192 (1961); 207.06435 (607)
Resveratrol-*O*-hexoside	389.1257	[M−H]^−^	4.946	−2.1	143.05177 (368); 157.06528 (442); 185.06099 (694); 227.07249 (6972)
Apigenin-*O*-hexoside	403.10266	[M+H]^+^, [M−H]^−^	4.3	−0.6	167.0338 (2511); 194.05806 (314); 222.05153 (614); 237.07681 (5309)
Astilbin	449.10883	[M−H]^−^	4.701	0.3	107.01384 (639); 151.00438 (1115); 151.03992 (566); 152.01263 (707)
Peonidin-*O*-hexoside Isomer I	461.10953	[M−2H]^−^	4.068	−2.5	147.009 (337); 148.017 (480); 229.05307 (137); 211.04135 (412); 227.03525 (582)
Peonidin-*O*-hexoside Isomer II	461.13187	[M−H]^−^	3.874	0.8	211.03944 (304); 227.03413 (353); 255.02911 (841); 256.03693 (524)
Quercetin-*O*-hexoside	463.08871	[M−H]^−^	4.591	−1.1	151.00304 (1045); 178.99954 (786); 255.02904 (2442); 300.02783 (13,213)
Delphinidin-*O*-hexoside	465.10376	[M]^+^, [M−2H]^−^	3.732	−0.7	116.9828 (268); 243.02551 (157); 285.03604 (149); 304.05365 (387)
Leucodelphinidin-*O*-hexoside	467.11896	[M−H]^−^	3.711	−0.9	109.02991 (617); 125.02531 (1108); 137.02364 (1114); 151.04021 (1024)
Epigallocatechin methylgallate	473.10883	[M+H]^+^	4.032	−2.0	255.06308 (316); 283.05536 (328); 311.05511 (12,324); 312.06018 (595)
Syringetin-*O*-hexoside	507.11331	[M−H]^−^	4.781	−1.0	242.02228 (705); 245.04523 (1696); 273.04019 (1122); 283.02597 (1040); 344.05469 (1393)
Dihydrodehydrodiconiferyl alcohol-*O*-hexoside	521.20343	[M−H]^−^	4.195	−1.1	109.03052 (1457); 313.10938 (2172); 344.12653 (7490); 359.15076 (3562)
Procyanidin B1 dimer 1	579.15118	[M+H]^+^	3.856	−1.5	123.04429 (2314); 127.03931 (7446); 135.04451 (2811); 139.03914 (4671)
Procyanidin B1 dimer 2	579.15094	[M+H]^+^	4.127	−0.9	123.04432 (3096); 127.0392 (5317); 135.04445 (1925); 139.03851 (3331)
Naringin	579.17334	[M−H]^−^, [M+FA−H]^−^, [M+H]^+^	4.647	−2.4	162.03224 (885); 177.0565 (3629); 205.05128 (1169); 219.06702 (2082)
Eriodictyol-*O*-hexoside-Pentoside	581.15173	[M−H]^−^	4.334	0.5	272.03314 (547); 300.02737 (3341); 315.05026 (3288); 316.05701 (529)
*O*-Coumaroyltrifolin	593.1297	[M−H]^−^	3.571	0.6	125.0249 (2188); 175.03925 (531); 177.01942 (741); 179.03432 (572)
Cyanidin-*O*-dihexoside	611.16071	[M]^+^	3.344	0.6	110.07099 (125); 221.08005 (155); 287.05588 (4460); 288.05411 (340)
Quercetin-*O*-dihexoside	625.14172	[M−H]^−^	4.295	−1.1	151.00378 (806); 271.02426 (941); 300.02713 (6067); 301.03458 (4671)
Procyanidin C1 dimer 1	865.19873	[M−H]^−^	4.286	−0.3	125.02456 (7189); 161.02429 (2838); 243.02972 (1783); 289.07251 (2062)
Procyanidin C1 dimer 2	865.19897	[M−H]^−^	4.03	0.1	125.02441 (3970); 161.02467 (2332); 243.02951 (1281); 407.07571 (1754)
Procyanidin C1 dimer 3	865.19916	[M−H]^−^	3.434	−0.1	125.02494 (3376); 243.03169 (1070); 289.07239 (1190); 407.07825 (1015)

**Table 3 foods-14-03857-t003:** Marker compounds of pitch coating in the samples, including VIP value and *p*-value.

Compound	VIP Value	*p*-Value
Coumaric acid	1.748	1.01 × 10^−6^
Esculetin	1.339	1.17 × 10^−4^
Tartaric acid	1.462	1.46 × 10^−12^
Procyanidin C1 dimer 1	1.270	1.21 × 10^−4^
Procyanidin C1 dimer 2	1.246	1.12 × 10^−4^
Procyanidin C1 dimer 3	1.228	8.42 × 10^−6^
Procyanidin B1 dimer 2	1.167	4.01 × 10^−4^
Kynurenic Acid	1.169	2.82 × 10^−8^

**Table 4 foods-14-03857-t004:** Anthocyanin concentrations in must and wine samples (µg/mL). Compounds with an asterisk (*) were putatively identified.

Sample	Delphinidin-3-*O*-glucoside * (µg/mL)	Cyanidin-3-*O*-glucoside (µg/mL)	Petunidin-3-*O*-glucoside (µg/mL)	Peonidin-3-*O*-glucoside * (µg/mL)	Malvidin-3-*O*-glucoside (µg/mL)
Must from non-coated vessel	8.27 ± 0.15	11.32 ± 0.21	18.11 ± 0.66	20.17 ± 0.21	60.58 ± 1.21
Must from coated vessel	7.08 ± 0.14	10.86 ± 0.25	16.03 ± 0.44	18.88 ± 0.50	57.29 ± 1.85
Wine from non-coated vessel	5.35 ± 0.33	5.10 ± 0.10	10.94 ± 0.42	9.10 ± 0.36	33.73 ± 0.87
Wine from coated vessel	7.96 ± 0.16	7.37 ± 0.08	15.05 ± 0.24	12.05 ± 0.17	39.92 ± 0.78
Difference *p*-value	0.0000	0.0002	0.0000	0.0005	0.0042

## Data Availability

The original contributions presented in this study are included in the article/Appendix A. Further inquiries can be directed to the corresponding authors.

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
