# Peer review of "Exploring the Impact of Pitch-Coated Pottery on Wine Composition: Metabolomics Characterization of an Ancient Technique"

_foods, 2025, doi:10.3390/foods14223857_

Round 1
Reviewer 1 Report
Comments and Suggestions for Authors
This study employs a foodomics approach to systematically characterize the effects of ancient pitch-coated pottery on wine fermentation and chemical composition. Through targeted and untargeted metabolomics analyses, it reveals significant impacts of the coating on key metabolites such as polyphenols and organic acids. The research demonstrates a rigorous design and advanced methodologies, providing a scientific basis for the modern application of traditional winemaking techniques. However, there remains room for enhancement in experimental details, data interpretation, and depth of conclusions.
Comments to the Author:
Question 1:The study demonstrates high scientific rigor through controlled experimental design (pitched vs. uncoated amphorae), triplicate sampling, and advanced analytical techniques (UPLC-MS/MS, GC-MS). However, fermentation conditions (temperature, duration, yeast source) are not specified, which they significantly influence metabolite profiles.
Question 2:The claim of “no migration of pitch compounds into wine” requires further validation (e.g., detection limit analysis for hazardous compounds like PAHs).
Question 3:The terminology clarification of “amphora” is valuable, but the historical accuracy could be strengthened by referencing archaeological studies on Roman dolia.
Question 4:The research objectives are clearly stated, but the rationale for focusing solely on Sangiovese grapes is not explained.
Question 5:Physical parameters of the pitch coating (thickness, uniformity, coverage) are not reported, making it difficult to assess actual contact area with wine.
Question 6:Sonication parameters for pitch extraction (25 min in ice bath) are justified, but temperature control during extraction requires more precise documentation.
Question 7:The UPLC-MS/MS methodology is robust, but the omission of internal standards for metabolite quantification may affect accuracy.
Question 8:The choice of extraction solvents (hexane vs. ethanol) lacks justification, and it is unclear how they cover the wide polarity range of compounds in pitch.
Question 9:Wine samples were diluted 1/10 and directly injected without assessing matrix effects on LC-MS quantification.
Question 10:Were the uncoated control vessels subjected to the same firing process? Was surface porosity measured? Otherwise, differences due to clay alone cannot be ruled out.
Question 11:The claim of being the “first study” on pitch's impact overlooks prior research on resin additives (e.g., mastic) in wine.
Question 12:The discussion fails to explain why pitch increases some polyphenols (e.g., anthocyanins) while decreasing others (e.g., procyanidins), lacking mechanistic insight.
Question 13:The discussion appropriately links reduced oxygen permeability in pitched vessels to polyphenol preservation, but overlooks potential microbial contributions to metabolic changes.
Question 14:The hypothesis that pitch coating redirects yeast metabolism is innovative but requires direct evidence (e.g., enzyme activity assays).
Question 15:Comparisons with prior studies (e.g., Tallah wines) are well-integrated, but differences in grape varieties and brewing methods should be discussed.
Question 16:Conclusions are consistent with the evidence but overstate the practical implications for modern oenology without addressing scalability.
Question 17:The conclusion that “pitch significantly influences wine composition” holds, but does not specify whether the effect is positive, negative, or neutral, lacking sensory or stability evaluation.
Question 18:The claim of “minimal compound migration” is supported by data, but the long-term stability of pitch coatings needs investigation.
Question 19:The conclusion fails to fully address the central question posed in the introduction: “How does pitch shape the fermentation environment and metabolite profile?” Mechanistic discussion is insufficient.
Author Response
Thank you very much for taking the time to review this manuscript. Please find the detailed responses below and the corresponding revisions highlighted in red in the re-submitted files. Following your suggestion, we have also carefully revised the English throughout the manuscript. In addition, we have improved the formatting of the tables and enhanced the resolution of Figure 1 to ensure better clarity.
Comment 1: The study demonstrates high scientific rigor through controlled experimental design (pitched vs. uncoated amphorae), triplicate sampling, and advanced analytical techniques (UPLC-MS/MS, GC-MS). However, fermentation conditions (temperature, duration, yeast source) are not specified, which they significantly influence metabolite profiles.
Response 1: We agree that parameters such as temperature, duration and yeast source can significantly influence metabolic profiles. To address this, we have revised Section 2.3 of the manuscript to provide clearer and more detailed information regarding the fermentation conditions. Specifically, we state that alcoholic fermentation was spontaneous, initiated by indigenous yeasts naturally present on the grape skins and/or in the environment, conducted at a controlled temperature of 25 °C, and lasted 10 days. It was performed with spontaneous fermentation following traditional procedures. These controlled and consistent conditions across all vessels were essential to ensure that any observed differences in metabolite profiles could be confidently attributed to the amphora coating rather than to variability in the fermentation process.
Comment 2: The claim of “no migration of pitch compounds into wine” requires further validation (e.g., detection limit analysis for hazardous compounds like PAHs).
Response 2: We thank the reviewer for highlighting this important aspect. The statement regarding the absence of migration of pitch-derived compounds into wine has been clarified in the revised manuscript (L385-393). Given that the study relied on a non-targeted metabolomics approach, it was not possible to establish specific detection limits for individual hazardous compounds such as PAHs. Therefore, although none of the potentially hazardous compounds identified in the pitch resin were detected in the wine samples, the possibility of trace-level migration cannot be entirely excluded without targeted analyses employing authentic standards. This limitation has now been explicitly stated in the manuscript to provide a more balanced interpretation of the results.
Comment 3: The terminology clarification of “amphora” is valuable, but the historical accuracy could be strengthened by referencing archaeological studies on Roman dolia.
Response 3: Thank you for your comment. We did not delve into the archaeological aspects due to limitations in terms of character count and references. We have clarified the use of dolia in Roman contexts (L-51-54). The references are scattered throughout book chapters and books, so we have chosen one of the most representative publications.
Comment 4: The research objectives are clearly stated, but the rationale for focusing solely on Sangiovese grapes is not explained.
Response 4: We have clarified the rationale in the revised manuscript (L128-130). Sangiovese was selected due to its historical and cultural significance in central Italy, particularly Tuscany, where it has been cultivated for centuries and is closely associated with traditional winemaking practices. The variety was first documented by Giovan Vettorio Soderini in 1590 as Sangiogheto, highlighting its longstanding role in Italian viticulture (Filippetti et al., 2005). Additionally, the grapes were locally available during the harvest period, allowing for controlled sourcing and traceability. Sangiovese is also a well-characterized variety in the scientific literature, making it a suitable model for fermentation studies.
Comment 5: Physical parameters of the pitch coating (thickness, uniformity, coverage) are not reported, making it difficult to assess actual contact area with wine.
Response 5: Thank you for your valuable comment. We have revised the manuscript (lines 121–122) to include additional details regarding the physical characteristics of the pitch coating. Specifically, we now report that the pitch layer applied to the ceramic vessels was approximately 2 mm thick, consistent with archaeological findings. The coating was applied evenly, resulting in a uniform layer across the entire internal surface of each vessel, ensuring full coverage and direct contact with the wine. These clarifications aim to better describe the contact conditions between the pitch and the wine during fermentation.
Comment 6: Sonication parameters for pitch extraction (25 min in ice bath) are justified, but temperature control during extraction requires more precise documentation.
Response 6: Thank you for your comments. We have revised the manuscript (lines 154–156) to provide more precise information regarding temperature control during sonication. The samples were fully submerged in an ice bath for the entire extraction period. While the temperature was not continuously monitored with a probe, the ice bath was replenished as needed to maintain a consistently cold environment, estimated to remain below 10 °C throughout the process.
Comment 7: The UPLC-MS/MS methodology is robust, but the omission of internal standards for metabolite quantification may affect accuracy.
Response 7: We thank the reviewer for this insightful comment. Internal standards were not included in the methodology, as the primary objective of the study was not the absolute quantification of metabolites, but rather a comparative analysis between different sampling areas. To ensure analytical robustness and monitor instrument performance, pooled quality control (QC) samples were injected regularly throughout the sequence. These QC samples enabled us to assess signal stability and mass accuracy (in ppm) across the dataset. As described in the manuscript, we evaluated the stability and reproducibility of the analysis by calculating the relative standard deviation (RSD%) of detected features in QC samples. The distribution of RSD% values is presented in Supplementary Figure S1, where most features exhibit low RSD%, supporting the robustness and consistency of the analytical platform. This clarification has now been included in the revised version of the manuscript (L181-183; L192-193).
Comment 8: The choice of extraction solvents (hexane vs. ethanol) lacks justification, and it is unclear how they cover the wide polarity range of compounds in pitch.
Response 8: We thank the reviewer for this constructive observation. Prior to selecting the extraction solvents, we conducted preliminary testing to evaluate which solvent would be most suitable for the chromatographic method and the analytical platform employed. Three solvents with different polarities—hexane, ethanol, and tert-butyl methyl ether (TBME)—were evaluated. Based on the results, hexane and ethanol provided the most effective and complementary extraction profiles, allowing for a broader coverage of pitch constituents. Hexane was particularly useful for extracting non-polar compounds, while ethanol enabled the recovery of more polar metabolites. This dual-solvent strategy allowed for a more exhaustive chemical characterization, which aligns well with the metabolite profiles reported in the literature. We have now clarified this the revised manuscript (L149-153).
Comment 9: Wine samples were diluted 1/10 and directly injected without assessing matrix effects on LC-MS quantification.
Response 9: Wine samples were diluted 1:10 prior to LC-MS analysis to reduce matrix complexity and minimize potential ion suppression effects. While a formal matrix effect assessment was not performed, this dilution strategy is widely accepted in non-targeted metabolomics workflows. As reported in literature (e.g., González-Centeno et al., 2020), many researchers analyse wine samples by simple filtration and dilution without performing polyphenol extraction, especially when the aim is to compare relative metabolite profiles.
Comment 10: Were the uncoated control vessels subjected to the same firing process? Was surface porosity measured? Otherwise, differences due to clay alone cannot be ruled out.
Response 10: We thank the reviewer for this observation. All amphorae used in the experiment, both pitch-coated and uncoated controls, were produced simultaneously using the same clay source and subjected to an identical firing process. This ensured consistency in manufacturing conditions and minimized variability due to ceramic composition or treatment. While surface porosity was not quantitatively measured, the vessels were handled and fired under the same conditions by the same producer, which supports the assumption of comparable baseline porosity. We acknowledge, however, that without direct porosity measurements, subtle differences cannot be entirely ruled out, and we have now clarified this limitation in the revised manuscript (L112-114).
Comment 11: The claim of being the “first study” on pitch's impact overlooks prior research on resin additives (e.g., mastic) in wine.
Response 11: We thank the reviewer for this valuable comment. We recognised that previous studies have investigated the use of resin additives such as mastic in winemaking and their impact on wine composition. Our intention was not to overlook this body of research, but rather to highlight that, to the best of our knowledge, this is the first study specifically focused on the chemical impact of pitch coating—as traditionally used to coat ceramic amphorae—on wine metabolite profiles using a non-targeted metabolomics approach and comparing coated and uncoated vessels. We have revised the manuscript to clarify this distinction and avoid any unintended generalization.
Comment 12: The discussion fails to explain why pitch increases some polyphenols (e.g., anthocyanins) while decreasing others (e.g., procyanidins), lacking mechanistic insight.
Comment 13: The discussion appropriately links reduced oxygen permeability in pitched vessels to polyphenol preservation but overlooks potential microbial contributions to metabolic changes.
Comment 14: The hypothesis that pitch coating redirects yeast metabolism is innovative but requires direct evidence (e.g., enzyme activity assays).
Response 12, 13, 14: We thank the reviewers for these insightful comments. The current study was conceived as an exploratory, non-targeted metabolomics investigation aimed at identifying differences in wine composition between pitch-coated and uncoated amphorae. While the data reveal significant changes in polyphenol profiles, such as increased anthocyanins and coumaric acid, and decreased procyanidins, and suggest possible influences of oxygen permeability and microbial dynamics, we acknowledge that the precise mechanisms remain to be elucidated.
The hypothesis that Pinaceae pine pitch coating may redirect yeast metabolism or modulate enzymatic activity is supported by metabolite trends, but direct evidence is currently lacking. As such, we consider this work a foundational step toward understanding how pitch affects the fermentation environment. Future studies involving targeted enzymatic assays, microbial community profiling, and transcriptomic analyses will be essential to confirm the biological and chemical mechanisms underlying the observed metabolite shifts.
We have clarified this limitation and the need for further mechanistic research in the revised manuscript (L455-457).
Comment 15: Comparisons with prior studies (e.g., Tallah wines) are well-integrated, but differences in grape varieties and brewing methods should be discussed.
Response 15: We thank the reviewer for this insightful comment. The manuscript has been revised to include a discussion of the differences in grape varieties and winemaking methods between our study and previous works, such as those related to Talha wines. These distinctions have been addressed throughout the manuscript where relevant.
Comment 16: Conclusions are consistent with the evidence but overstate the practical implications for modern oenology without addressing scalability
Response 16: We appreciate the reviewer’s observation. In response, we have revised the conclusions to adopt a more cautious tone regarding the practical implications of our findings for modern oenology. Specifically, we now clarify that any potential applications should be considered exploratory and that further research is needed to assess scalability, reproducibility, and feasibility in contemporary production settings (L497-506). This change aims to better align the conclusions with the scope and limitations of the study.
Comment 17: The conclusion that “pitch significantly influences wine composition” holds, but does not specify whether the effect is positive, negative, or neutral, lacking sensory or stability evaluation.
Response 17: Thank you for this valuable comment. We agree that the original conclusion lacked specificity regarding the nature of the compositional changes. We have now clarified that while pitch coating significantly influences wine composition, the direction of these changes, whether positive, negative, or neutral, remains to be determined (L497-506). As sensory and stability evaluations were beyond the scope of this study, we have acknowledged the need for future work to assess the organoleptic and quality-related implications of the observed metabolomic shifts.
Comment 18: The claim of “minimal compound migration” is supported by data, but the long-term stability of pitch coatings needs investigation.
Response 18: Thank for this consideration. Our study focused exclusively on the alcoholic fermentation (AF) stage, which typically lasts 5-10 days. Under these conditions, the claim of “minimal compound migration” is supported by our analytical data. However, we acknowledge that longer-term contact, such as during wine ageing, may present different migration dynamics. A dedicated study on the long-term stability of pitch coatings would be necessary to fully assess their behavior over extended periods. This limitation has now been clarified in the revised manuscript (lines 391–393).
Comment 19: The conclusion fails to fully address the central question posed in the introduction: “How does pitch shape the fermentation environment and metabolite profile?” Mechanistic discussion is insufficient.
Response 19: We thank the reviewer for this valuable observation. In response, we have revised the aim of the study in the Introduction to more clearly reflect the central question evaluate the impact of pitch-coated amphorae on wine composition through a comparative, non-targeted metabolomics approach, acknowledging the exploratory nature of the work (L86-87).
While our findings reveal significant differences in metabolite profiles between wines fermented in coated and uncoated vessels, we recognize that the mechanistic basis of these changes, whether due to oxygen permeability, microbial dynamics, or yeast metabolism, requires further investigation. We have clarified in the revised manuscript that this study serves as a pilot study, and future research involving targeted assays and microbial profiling will be necessary to fully elucidate the mechanisms involved.
Reviewer 2 Report
Comments and Suggestions for Authors
- 99-103 But what was it produced from?
- 110-117 Was the coating uniform?
- 131-133 How was the wine collected? Was it strained, filtered, decanted, centrifuged or collected with grape residue and particles?
- 140 At what temperature?
- 142 amber-glass or amber-coloured, not amber.
- 221 What % similarity was determined as ‘identified’?
- 238 Which were available and which were unavailable? Perhaps could you note it in the table with the compounds
- 282 Italics for Latin.
- 348-349, Table 2 – text says that 69 compounds were identified in must and wine. Table says that it was identified in wine. No information is given whether these compounds were identified in all of the samples, or some were identified only in wine, pitch-wine, must or not. Please, clarify that information.
- 356-358 – But the amphorae without the pitch was used, so you can tell, whether it originated in resin or not, right?
Author Response
Thank you very much for taking the time to review this manuscript. Please find the detailed responses below and the corresponding revisions highlighted in red in the re-submitted files.
Comment 1: 99-103 But what was it produced from?
Response 1: Thank you for your input, we clarified that pitch was produced from resin-rich pieces of Pinaceae wood (L102-103).
Comment 2: 110-117 Was the coating uniform?
Response 2: Thank you for your comment. We have addressed this point by adding a clarification in lines 121–122 of the revised manuscript, indicating that the pitch coating was applied evenly across the internal surface of the vessels.
Comment 3: 131-133 How was the wine collected? Was it strained, filtered, decanted, centrifuged or collected with grape residue and particles?
Response 3: Thank you for your question. We have clarified this point in lines 140–144 of the revised manuscript. Specifically, we now indicate that samples were collected at both the beginning and the end of alcoholic fermentation (AF), and in both cases, samples were filtered through a stainless-steel mesh to remove grape residues and suspended particles prior to collection. All samples were stored in an ultra-freezer at –60 °C until chemical analyses were performed.
Comment 4: 140 At what temperature?
Response 4: Thank you for your comments. We have revised the manuscript (lines 154–156) to provide more precise information regarding temperature control during sonication. The samples were fully submerged in an ice bath for the entire 25-minute extraction period. While the temperature was not continuously monitored with a probe, the ice bath was replenished as needed to maintain a consistently cold environment, estimated to remain below 10 °C throughout the process.
Comment 5: 142 amber-glass or amber-coloured, not amber.
Response 5: The reviewer was right. We have corrected the terminology in the revised manuscript, replacing “amber” with “amber-glass” in lines 159 and 169 to ensure clarity and accuracy.
Comment 6: 221 What % similarity was determined as ‘identified’?
Response 6: Thank you for your question. We have clarified this point in the lines L207-208. In MS-DIAL, compounds were considered putatively identified (level 2) when the total spectral similarity score exceeded 75%, based on mass accuracy, isotopic pattern, and MS/MS spectral matching. This threshold aligns with commonly accepted standards in untargeted metabolomics workflows and ensures a reliable level of annotation without requiring reference standards
Comment 7: 238 Which were available and which were unavailable? Perhaps could you note it in the table with the compounds
Response 7: Thank you for your suggestion. We have clarified this point in the revised manuscript (L264) and updated Table 4 accordingly. Compounds for which reference standards were unavailable and thus putatively identified (level 2) are now marked with an asterisk (*).
Comment 8: 282 Italics for Latin.
Response 8: Thank you for your careful reading. We have corrected the formatting by italicizing Latin terms as appropriate in line 305.
Comment 9: 348-349, Table 2 – text says that 69 compounds were identified in must and wine. Table says that it was identified in wine. No information is given whether these compounds were identified in all of the samples, or some were identified only in wine, pitch-wine, must or not. Please, clarify that information.
Response 9: Thank you for your observation. We have corrected the heading of Table 2 to more accurately reflect the content and clarified in the manuscript (L375-378) that the full distribution of compounds across sample types (must, wine from pitch-coated vessels, and wine from uncoated vessels) is provided in Supplementary Material 2, which includes the peak area data for all samples. This allows readers to assess in which samples each compound was detected.
Comment 10: 356-358 – But the amphorae without the pitch was used, so you can tell, whether it originated in resin or not, right?
Response 10: We agree that the use of uncoated amphorae allows for comparison and helps assess whether ethyl vanillin originated from the pitch coating. As noted in the revised manuscript, ethyl vanillin was detected in both pitch-coated and uncoated vessels, suggesting that its presence is more likely due to fermentation rather than leaching from the resin. We have clarified this point in the text (L383-384) to reflect the comparative value of the control vessels.
Round 2
Reviewer 1 Report
Comments and Suggestions for Authors
Q1: There is too much blank space between pages 15 and 16.
Q2: The format of the references is not standardized.
Comments on the Quality of English LanguageQ1: There is too much blank space between pages 15 and 16.
Q2: The format of the references is not standardized.